# Elevated ROS Levels Caused by Reductions in GSH and AsA Contents Lead to Grain Yield Reduction in Qingke under Continuous Cropping

**DOI:** 10.3390/plants13071003

**Published:** 2024-03-31

**Authors:** Xue Gao, Jianxin Tan, Kaige Yi, Baogang Lin, Pengfei Hao, Tao Jin, Shuijin Hua

**Affiliations:** 1State Key Laboratory of Hulless Barley and Yak Germplasm Resources and Genetic Improvement, Lhasa 850002, China; gaoxue365@163.com (X.G.); m18898010927@163.com (J.T.); 11816004@zju.edu.cn (P.H.); 2Institute of Crop and Nuclear Technology Utilization, Zhejiang Academy of Agricultural Sciences, Hangzhou 310021, China; kaige_y0113@163.com (K.Y.); linbg@mail.zaas.ac.cn (B.L.)

**Keywords:** ascorbic acid, glutathione, lipid peroxidation, Qingke, redox, reactive oxygen species, yield

## Abstract

Continuous spring cropping of Qingke (*Hordeum viilgare* L. var. nudum Hook. f.) results in a reduction in grain yield in the Xizang autonomous region. However, knowledge on the influence of continuous cropping on grain yield caused by reactive oxygen species (ROS)-induced stress remains scarce. A systematic comparison of the antioxidant defensive profile at seedling, tillering, jointing, flowering, and filling stages (T1 to T5) of Qingke was conducted based on a field experiment including 23-year continuous cropping (23y-CC) and control (the first year planted) treatments. The results reveal that the grain yield and superoxide anion (SOA) level under 23y-CC were significantly decreased (by 38.67% and 36.47%), when compared to the control. The hydrogen peroxide content under 23y-CC was 8.69% higher on average than under the control in the early growth stages. The higher ROS level under 23y-CC resulted in membrane lipid peroxidation (LPO) and accumulation of malondialdehyde (MDA) at later stages, with an average increment of 29.67% and 3.77 times higher than that in control plants. Qingke plants accumulated more hydrogen peroxide at early developmental stages due to the partial conversion of SOA by glutathione (GSH) and superoxide dismutase (SOD) and other production pathways, such as the glucose oxidase (GOD) and polyamine oxidase (PAO) pathways. The reduced regeneration ability due to the high oxidized glutathione (GSSG) to GSH ratio resulted in GSH deficiency while the reduction in L-galactono-1,4-lactone dehydrogenase (GalLDH) activity in the AsA biosynthesis pathway, higher enzymatic activities (including ascorbate peroxidase, APX; and ascorbate oxidase, AAO), and lower activities of monodehydroascorbate reductase (MDHAR) all led to a lower AsA content under continuous cropping. The lower antioxidant capacity due to lower contents of antioxidants such as flavonoids and tannins, detected through both physiological measurement and metabolomics analysis, further deteriorated the growth of Qingke through ROS stress under continuous cropping. Our results provide new insights into the manner in which ROS stress regulates grain yield in the context of continuous Qingke cropping.

## 1. Introduction

Reactive oxygen species (ROS) mainly include hydrogen peroxide (H_2_O_2_), superoxide anions (O_2_^•−^), and hydroxyl radicals (•OH), which are characterized as oxygen-containing species with oxidative reactivity [1]. The production of ROS often occurs in aerobic cells during the electron transport chain reaction in mitochondria [2]. Investigations have shown that ROS take part in many plant metabolic processes, such as intracellular transcription factor activation, gene expression, and cell proliferation [3,4,5]. Although excessive ROS causes cell damage and even death [6], accumulated evidence has shown that some ROS, such as hydrogen peroxide, can enhance plant growth and development when in small amounts [7,8,9]. For example, hydrogen peroxide can activate flavonoid biosynthesis, ethylene synthesis, and the signaling pathway to promote K^+^ absorption through modulating K^+^ transporter genes in grapevines [10]. On the other hand, results from many previous investigations have demonstrated that excessive ROS causes diseases in plants [11,12]. In summary, as important molecules, ROS exist in plants, are involved in many types of physiological metabolism, and play important roles in the regulation of plant growth and development.

The damage that excessive ROS causes to cell membranes can be repaired to different extents, depending on the ROS scavenging ability. There exist several systems to quench excessive ROS, including the glutathione (GSH) system, the ascorbic acid (AsA) system, and other enzymatic systems such as catalase (CAT) and superoxide dismutase (SOD) [13]. SOD functions as the first defense to decompose superoxide anion (SOA) radicals and catalyzes O_2_^•−^ to H_2_O_2_ [14,15]. Further, CAT and glutathione peroxidase (GPX) reduce H_2_O_2_ to H_2_O [16]. Those antioxidant enzymes have a ubiquitous role in mitigating different abiotic stresses including pesticides, heavy metals, drought, etc., which generate ROS [17]. In addition to those enzymes, glutathione is an essential antioxidant and low-molecular weight thiol that maintains cellular redox balance, which is composed of glutamate, cysteine, and glycine [18]. As a free radical scavenger, glutathione possesses an active thiol group, which is easily oxidized and dehydrogenated to eliminate O_2_^•−^ providing electrons, allowing some enzymes (including GPX) to reduce H_2_O_2_ to H_2_O [19]. In addition to glutathione, ascorbic acid (AsA) is an important antioxidant compound, being a non-enzymatic molecule that detoxifies ROS [20]. At present, the exact regulatory network of ROS removal by AsA has still not been sufficiently elucidated. However, many investigations have shown that AsA can reduce the ROS level to prevent damage caused by ROS, thus enhancing plant yield and quality [21,22]. For example, the exogenous application of AsA to fresh-cut potatoes inhibited wound healing by reducing H_2_O_2_ and O_2_^•−^, increasing enzymes such as CAT and GPX, and maintaining high levels of AsA and GSH [23]. Liu et al. (2023) found that, when kiwifruit was stressed with Cu for 12 h, the AsA content in leaves increased and key genes involving AsA biosynthesis—namely, *GDP-*L*-galactose phosphorylase3* (*GGP3*) and *GDP-mannose-3′,5′-epimerase* (*GME*)—were upregulated [24]. Furthermore, the over-expression of *GGP3* increased the endogenous AsA content and alleviated Cu toxicity through decreasing ROS levels [24]. In conclusion, the relationship between enzymatic and non-enzymatic antioxidants and ROS is a key factor for redox homeostasis affecting plant growth and development.

Crops, including Qingke, often suffer from ROS stress under adverse environments such as drought and low temperature, as well as improper agronomic practices including continuous cropping. Qingke is the most important food crop for people in the Qinghai–Xizang Plateau. Qingke grows in environments with the characteristics of low temperature, low oxygen, and low nutrients in the soil. Therefore, Qingke is normally grown in spring (late March to early April) and harvested in autumn (August), and the land is left in a fallow status for the months after harvesting. However, long-term monoculture of Qingke results in continuous cropping obstacles, with effects such as slowed growth and reductions in grain yield. The inhibition of growth in Qingke may be affected by many factors, including ROS stress, during continuous cropping. It has been reported that the level of ROS and the expression of *peroxidase* in roots of continuous-cropping strawberries were higher than those in non-continuous cropping roots through genome-wide analysis of the transcription WRKY family and a transcriptomic study [25]. When *Angelica sinensis* was under continuous cropping, a decrease in antioxidant enzyme activities (including CAT, SOD, and POD) partly explained the yield reduction [26]. Similar results have also been reported in different plant species, including reductions in antioxidant enzymes and increases in ROS levels and malondialdehyde (MDA) content [27,28,29]. However, investigations on the impacts of continuous Qingke cropping on ROS profile status have not been reported, thus confining crop stress-related breeding and cultivation. 

Therefore, we hypothesized that the antioxidant system would be greatly affected by continuous cropping at a physiological level, which is helpful in providing evidence for further optimization of agronomic practices for Qingke production. In the current study, the level of ROS from Qingke plants undergoing the 23y-CC treatment at different developmental stages was analyzed first, in order to ensure whether they were elevated when compared to the control. Then, enzymatic and non-enzymatic systems were monitored to understand whether the produced ROS can be scavenged in a timely manner. Finally, samples from 23y-CC and the control were also assayed through metabolomics analysis to confirm the ROS production and maintenance of appropriate redox status in cells. To the best of our knowledge, this is the first report of a systematic investigation of the effect of continuous cropping on the redox regulation of Qingke, which is beneficial to readjusting the planting mode for Qingke production in Xizang.

## 2. Results

### 2.1. Yield Performance and ROS Production in Qingke Leaf under 23-Year Continuous Cropping (23y-CC) at Different Developmental Stages

The grain yield of Qingke under the control was significantly higher than that of the 23y-CC treatment, which decreased by 38.67% (Figure 1a). 

Superoxide anion content (SOA) showed an increasing trend from T1 to T5, both under 23y-CC and the control (Figure 1b). The SOA content at T5 increased by 2.14 and 2.19 times, when compared to T1 under 23y-CC and the control, respectively. Furthermore, the SOA content from T1 to T5 under the 23y-CC treatment was significantly higher than control at each developmental stage. In the T2 and T3 stages, SOA content under the 23y-CC treatment was 1.87 and 1.74 times higher than those in the control (Figure 1b). 

Regarding hydrogen peroxide content, both 23y-CC and the control first increased and then decreased, but with different inflection points. For the 23y-CC treatment, the hydrogen peroxide content peaked at T2, then decreased slightly from T2 to T3 and decreased sharply after; in particular, the hydrogen peroxide content at T5 was decreased by 44.39% compared to T2. For the control, the hydrogen peroxide content reached a maximum at T3 and then decreased quickly: the content at T5 was decreased by 47.53% compared to T3. In the T1 and T2 stages, the hydrogen peroxide content under the 23y-CC treatment was significantly higher than that under the control, while a reverse trend was found for the T4 stage. It was observed that the largest difference in hydrogen peroxide content between 23y-CC and the control was at T1, the seedling stage, in which the content under 23y-CC treatment was 10.41% higher than that for the control (Figure 1c).

Membrane lipid peroxidation (LPO) changed differently between the 23y-CC and control treatments at different Qingke development stages. For the 23y-CC treatment, LPO slightly decreased from T1 to T2 and then increased rapidly from T2 to T3. However, the LPO stayed relatively stable from T3 to T5. For control, LPO increased from T1 to T2 greatly and then showed a decreasing trend to T5. There was no significant difference between treatments at the T1 stage. However, in the T2 stage, LPO under the 23y-CC treatment was considerably higher than that of the control, being 2.61-fold higher than the control. From T3, LPO under 23y-CC was significantly higher than that under the control. In the T4 and T5 stages, LPO under 23y-CC was 44.36% and 44.60% higher than that of the control, respectively (Figure 1d).

Malondialdehyde (MDA) is one of the degradative products during lipid peroxidation, which is an important index of the LPO level. The results showed that the MDA content under the 23y-CC treatment was significantly higher than that under the control, while no significant differences were found in the T1 and T2 stages. The MDA content under the control from T3 to T5 decreased by 73.03%, 63.56%, and 100.00%, respectively, when compared to the 23y-CC treatment. The MDA content generally showed a decreasing trend from T1 to T5 under the control, but increased from T1 to T3 and then decreased under the 23y-CC treatment. The MDA content in the T5 stage decreased by 64.32% compared to T3 under the 23-CC treatment (Figure 1e). 

Nitrotetrazolium blue chloride (NBT) staining was performed, and the SOA content obviously accumulated more under 23y-CC than under the control at different developmental stages (Figure 1f). The detection of hydrogen peroxide content through diaminobenzidine staining generally agreed with physiological measurement results (Figure 1g). The above results suggest that the increased ROS levels under 23y-CC resulted in a high LPO level.

### 2.2. ROS Scavenging Ability

Keeping the redox status in balance through timely ROS scavenging is a vital factor for healthy Qingke growth under the ROS stress caused by continuous cropping. The results of the hydroxyl radical scavenging ability (HRSA) test showed that the HRSA under the 23y-CC treatment was lower than that for the control at each developmental stage. However, in the T2 and T4 stages, the differences in HRSA between the 23-CC treatment and control were not significant. In the T1, T3, and T5 stages, the HRSA under the control was 84.11%, 35.96%, and 99.12% higher than that under the 23y-CC treatment, respectively. As for different developmental stages, the HRSA started increasing at T1 and peaked at T3, increasing by 40.10%, and then decreased under the control. The HRSA had a generally increasing trend and peaked at T4, increasing by 145.79% compared to the T1 stage, and then decreased to a similar level to T1 at the T5 stage under the 23y-CC treatment (Figure 2a). 

The superoxide anion scavenging capacity (SASC) was significantly higher under 23y-CC than the control from Stages T2 to T4, while the reverse trend was observed for the T1 and T5 stages. However, there was no significant difference between the 23y-CC treatment and the control in the T5 stage. From Stages T2 to T4, the SASC under 23y-CC was 25.05%, 17.74%, and 51.25% higher than that for the control, respectively. For different developmental stages, the SASC slightly increased from T1 to T3 and then decreased sharply under the 23y-CC treatment. The SASC at T5 decreased by 57.59% compared with T3 under the 23y-CC treatment. For the control, the SASC showed a decreasing trend from T1 to T5 overall, but with a slight increase from Stage T2 to T3. The SASC at T5 decreased by 51.04% compared with T1 under the control (Figure 2b).

The total antioxidant capacity (T-AOC) under the 2,2-Diphenyl-1-picrylhydrazyl (DPPH) method significantly decreased under the 23y-CC treatment as compared with the control at all developmental stages. The biggest difference between 23y-CC and the control was at the T1 stage, reaching a 2.08-fold difference. For different developmental stages, both treatments showed the greatest capacity at the T1 stage and then decreased. However, the decreasing rate from T1 to T2 was much smaller under the 23y-CC treatment than that of the control, which exhibited decrements of 49.12% and 20.95%, respectively. From T2 to T5, the capacity was kept relatively stable for the 23y-CC treatment but markedly increased from T4 to T5 under the control (Figure 2c). 

### 2.3. ROS Scavenging Systems

#### 2.3.1. Glutathione (GSH) System

Reduced glutathione (GSH) is an important antioxidant in plants. The results showed that GSH content under the 23y-CC treatment was significantly lower than that under the control from T1 to T4, with no significant difference observed in the T5 stage. The biggest difference between the 23y-CC treatment and the control was in T2, which was 1.48-fold. For different developmental stages, a stable GSH content from T1 to T3 was observed under the control; however, an increase in GSH content was observed, resulting in a peak at T4. For the 23y-CC treatment, the GSH content decreased from T1 to T2 and then quickly increased from T2 to T3. The GSH content also peaked at T4 under the 23y-CC treatment. However, the GSH content at T5 and T2 decreased by 25.54% and 39.74% compared to T4 under the control and 23y-CC treatments, respectively (Figure 3a).

Oxidized glutathione (GSSG) content was drastically lower than GSH content under continuous cropping treatment for all developmental stages. There was no significant difference in GSSG content at the T1 and T5 stages. The GSSG content under the control was significantly higher than that under the 23y-CC treatment for Stages T2 to T4. The GSSG content under the 23y-CC treatment decreased by 70.47%, 75.86%, and 95.77% at these stages compared to the control, respectively. The GSSG content showed a decreasing trend from T1 to T4 under the 23y-CC treatment, while it increased substantially from T1 to T2 and then decreased sharply from T2 to T5 under the control. The GSSG content at T2 and T1 decreased by 97.45% and 94.25% compared to T5 under the control and 23y-CC treatments, respectively (Figure 3b).

Glutathione peroxide (GPX) activity was significantly higher under 23y-CC than the control before the T3 stage, while the reverse trend was observed at Stages T4 and T5. At Stages T1 and T2, the GPX activity under the control was 45.23% and 35.13% lower than that of the 23y-CC treatment, respectively (Figure 3c). For the difference in developmental stages, the GPX activity exhibited a decreasing trend from the T1 to T5 stage under the 23y-CC treatment, while the activity increased first, peaked at T3, and then decreased again. The GPX activity at the T5 stage decreased by 41.67% and 22.16% compared to Stages T1 and T3 under the 23y-CC and control treatments, respectively (Figure 3c).

Glutathione S-transferase (GST) activity was lower under the 23y-CC treatment than under the control, but their differences were not significant for Stages T1 and T2. However, the GST activity from Stages T3 to T5 under the 23y-CC treatment was significantly higher than under the control except for T4, in which the difference was not significant. In Stages T3 and T5, the GST activity under the 23y-CC treatment was 97.78% and 30.60% higher than that under the control, respectively. Both the control and 23y-CC treatments showed a quick increase from T1 to T2 and then decreased to minimum activity at Stages T3 and T4, respectively. A slight recovery of activity was observed under both treatments. The GST activity in the T3 and T4 stages was decreased by 71.20% and 57.42% compared to the T2 stage, respectively (Figure 3d).

Glutathione reductase (GR) activity was significantly higher under the control compared to the 23y-CC treatment at the T1 and T2 stages, while the reverse trend was observed for the T3 stage. The GR activity under the 23y-CC treatment significantly decreased by 44.34% and 55.69% compared to the control at Stages T1 and T2, respectively. However, in the T3 stage, GR activity under the control decreased by 83.79% compared to the 23y-CC treatment. The GR activity under the control increased slightly from T1 to T2 and then sharply decreased from T2 to T3. The decrement in GR activity from T2 to T3 reached 94.44% under the treatment. For the 23y-CC treatment, a decreasing trend of GR activity was observed; in particular, the most rapid decrease was from T3 to T4, which reached 80.17% (Figure 3e).

For thioredoxin peroxidase (TPX) activity, significant differences were found at Stages T2 and T4. The TPX activity at the T2 stage under the 23y-CC treatment was decreased by 26.68% compared to the control. However, the TPX activity at the T4 stage under the control was decreased by 67.44% compared to the 23y-CC treatment. Regarding the influence of the developmental stage on TPX activity, both treatments showed a decreasing trend. The TPX activity at the T5 stage was lowered by 86.19% and 91.38% when compared to the T1 stage under the control and 23y-CC treatments, respectively (Figure 3f).

Glutamatecysteineligase (GCL) activities under the 23y-CC treatment from T1 to T3 were relatively stable and very high, then dropped to nearly zero. However, the GCL activity under the control was very low throughout the developmental stages. The GCL activity at Stages T1 to T3 under the control was decreased by 87.87%, 93.86%, and 94.16% compared to those under the 23y-CC treatment, respectively (Figure 3g). 

Thioredoxin reductase (TrxR) activity under the 23y-CC treatment was significantly higher than that of the control from T1 to T3. The TrxR activity under the 23y-CC treatment decreased from T1 to T3 by 76.89%, 83.69%, and 76.29% compared to the control, respectively. The TrxR activity under the control rapidly increased from T1 to T2 and then sharply decreased from T2 to T4. The TrxR activity at T4 was decreased by 96.19%, compared to T2, under the control. However, the TrxR activity under 23y-CC was very low during Qingke development (Figure 3h). 

#### 2.3.2. AsA System

Ascorbic acid (AsA) plays important roles in eliminating radicals. The results showed that the AsA contents in Stages T1 and T2 under the control were 1.46- and 2.03-fold higher than those under the 23y-CC treatment, while the AsA content at the T4 stage under the 23y-CC treatment was 1.23-fold higher than that under the control. There was no significant difference between the control and 23y-CC treatments in Stages T3 and T5. Both treatments showed maximum contents in Stage T3. The AsA content in Stage T5 was decreased by 33.72% and 34.58% compared to Stage T3, respectively, under the control and 23y-CC treatments (Figure 4a).

Dehydroascorbate (DHA) is a reversible product of oxidized AsA. The results showed that the DHA content under the 23y-CC treatment was significantly higher than that under the control at each developmental stage except T4. The DHA content under the control was decreased by 21.41%, 18.23%, 17.20%, and 20.03% compared to the 23y-CC treatment in Stages T1, T2, T3, and T5, respectively. The DHA content generally exhibited a decreasing trend under both treatments. The DHA contents under the control and 23y-CC treatments at the T5 stage were decreased by 28.45% and 29.68%, respectively (Figure 4b).

Dehydroascorbate reductase (DHAR) functions in the reduction reaction between GSH and DHA as their catalyzer and produces AsA and GSSG. The results showed that the DHAR activity under the control treatment in T2, T3, and T5 was significantly higher than that under the 23y-CC treatment, which presented decrements of 15.22%, 33.71%, and 11.61%, respectively. In the T1 stage, the DHAR activity under the 23y-CC treatment was significantly higher than that under the control, and no significant difference was observed between the two treatments at Stage T3. Under the control, the DHAR activity increased from T1 and peaked at T4, while the maximum activity was observed in the T3 stage under the 23y-CC treatment. The DHAR activity at T5 decreased by 29.61% and 40.76% compared to Stages T4 and T3 under the control and 23y-CC treatments, respectively (Figure 4c).

Ascorbate peroxidase (APX) catalyzes the reaction of hydrogen peroxide oxide AsA and is an important scavenger of hydrogen peroxide and a key enzyme in AsA metabolism. The results showed that the APX activity under the 23y-CC treatment from T1 to T3 was significantly higher than that under the control, which exhibited decrements of 69.49%, 66.92%, and 67.20%, respectively. Both treatments showed a decreasing trend of APX activity from T1 to T5. The APX activity at T5 was decreased by 77.69% and 100% compared to T1 under the control and 23y-CC treatments, respectively (Figure 4d). 

Monodehydroasorbate reductase (MDHAR) catalyzes MDHA by reducing it to AsA, which plays an important role in regulating redox in AsA metabolism. The results revealed a significantly higher MAHAR activity under the control throughout the developmental stages, when compared to the 23y-CC treatment. The MDHAR activity under the 23y-CC treatment was decreased by 67.35%, 73.64%, 45.04%, 76.22%, and 77.40% as compared to the control from T1 to T5, respectively. The MDHAR activity under the 23y-CC treatment was relatively stable at each developmental stage. However, for the control, the activity in Stages T1, T2, and T4 showed few variations, while being significantly reduced in T3 and T5 (Figure 4e). 

Ascorbate oxidase (AAO) can direct oxidize AsA, thus regulating plant cell wall metabolism. The results showed that the AAO activity under the 23y-CC treatment was higher than that under the control at Stages T1 and T2. However, the AAO activity under the 23y-CC treatment from the T3 stage was significantly higher than that under the control. The AAO activity under the control was decreased by 75.20%, 65.44%, and 64.89% compared to the 23y-CC treatment from T3 to T5, respectively. The AAO activity showed very few variations from T1 to T2. However, reverse trends were found between the treatments. The AAO activity under the control significantly decreased from T2 to T3 and was then kept relatively stable, while it significantly increased from T2 to T3 and then remained stable as well under 23y-CC. The AAO activity from T2 to T3 decreased/increased by 63.82% and 48.65% under the control and 23y-CC treatments, respectively (Figure 4f). 

The L-galactono-1,4-lactone dehydrogenase (GalLDH) pathway is the main biosynthetic pathway of AsA, which is responsible for the last step of AsA biosynthesis in plants. The results showed that GalLDH activity under the 23y-CC treatment was significantly lower than that under the control from T2 onward. The GalLDH activity under the 23y-CC treatment from T2 was decreased by 19.84%, 39.69%, 58.08%, and 59.71% compared to the control, respectively. The GalLDH activity from T1 to T2 increased and then decreased after T2 under both treatments. However, the GalLDH activity after T2 sharply decreased to T4 under the 23y-CC treatment, while that from T3 to T4 decreased very quickly under the control. The GalLDH activity at Stage T5 was decreased by 92.18% and 96.07% under the control and 23-CC treatments, respectively (Figure 4g).

#### 2.3.3. Enzymatic System

Catalase (CAT) is the most important enzyme for the removal of hydrogen peroxide. The results showed that CAT activity under the 23y-CC treatment in Stages T1 and T2 was higher than that under the control, while a reverse trend was observed for the T3 stage. CAT activity under the 23y-CC treatment was decreased by 69.98%, 87.71%, and 90.71% compared to the control from T3, respectively. The CAT activity first decreased linearly from T1 to T4, while the activity decreased from T1 to T2, increased from T2 to T3, and then decreased again to the level at T2 under the control. The CAT activity at T5 was decreased by 77.53% and 97.58% compared to T3 and T1 under the control and 23y-CC treatments, respectively (Figure 5a).

Peroxidase (POD) exists extensively in animal, plants, and other species, and catalyzes hydrogen peroxide to eliminate its toxicity in plant tissues. The results showed that POD activity under the 23y-CC treatment was significantly higher than that under the control at Stages T2 and T3, while the reverse trend was observed for the T4 stage. The POD activity under the control at T2 and T3 decreased by 41.30% and 57.40% compared to the 23y-CC treatment. The POD activity under the 23y-CC treatment at the T4 stage was decreased by 43.79% compared to the control. The POD activity increased from T1 to T2, then decreased at the T3 stage; however, it increased again and reached a maximum value at T4 under the control. However, the POD activity increased from T1 to T2, kept its activity at T3, and then decreased sharply almost to the T1 level at T4 and T5 under 23y-CC. The maximum POD activities at the T4 and T3 stages were 1.08- and 2.60-fold higher than that at the T5 stage under the control and 23y-CC treatments, respectively (Figure 5b).

Superoxide dismutase (SOD) catalyzes the disproportionate reaction of superoxide anions to produce hydrogen peroxide, which plays an important role in the plant antioxidant system. The results showed that the SOD activity under 23y-CC was significantly lower than that under the control, except in the T1 stage. The SOD activity under the control was decreased by 54.68%, 31.97%, 69.54%, and 87.40% from Stages T2 to T5 compared to the 23y-CC treatment. The SOD activity under the control from T1 to T2 showed a slight increase, then decreased sharply and remained at a relatively stable level. However, the SOD activity exhibited a decreasing trend under 23y-CC. The SOD activity at the T5 stage was decreased by 57.68% and 95.07%, compared to the T1 stage, under the control and 23y-CC treatments, respectively (Figure 5c).

Glucose oxidase (GOD) catalyzes glucose to be oxidized into gluconic acid and hydrogen peroxide. The results showed that GOD activity under the 23y-CC treatment was significantly higher than that under the control in Stages T2 and T3, while the reverse trend was observed for the other stages. The biggest difference in GOD activity appeared at T3, which had a 3.22-fold difference. The GOD activity under the control was relatively stable from T1 to T3, peaked at T4, and then decreased again. For the 23y-CC treatment, the GOD activity considerably increased from T1 to T3 and then decreased quickly. The GOD activity in the T5 stage was decreased by 62.45% and 89.58% compared to Stages T4 and T3 under the control and 23y-CC treatments, respectively (Figure 5d).

Polyamine oxidase (PAO) catalyzes polyamine to be oxidized to produce hydrogen peroxide. The results showed that PAO activity under the 23y-CC treatment was significantly higher than that under the control from T2 to T4, while a reverse trend was found for the T1 stage. The PAO activity under the control at T2 to T4 was decreased by 29.72%, 44.07%, and 79.03% compared to the 23y-CC treatment, respectively. Both treatments showed a peak of PAO activity at T2, followed by a rapid decrease. The PAO activity at T5 decreased by 92.74% and 94.83%, compared to T2, under the control and 23-CC treatments, respectively (Figure 5e).

Diamine oxidase (DAO) catalyzes diamine to be oxidized into aldehydes, which produces hydrogen peroxide and is closely associated with nucleic and protein biosynthesis. The results showed that the DAO activity under the 23y-CC treatment was significantly higher than that under the control from the T1 to T4 stages. The differences in DAO activity between treatments were 1.89-,1.42-, 1.94-,4.63-, and 2.06-fold from T1 to T5, respectively. The DAO activity peaked at T2 and then decreased linearly under both treatments, and the DAO activity at T5 was decreased by 92.14% and 94.59% compared to T2, respectively (Figure 5f).

### 2.4. Alterations of Other Antioxidants under Different Continuous Cropping Treatments

Flavonoids are a class of polyphenyl secondary compounds that can eliminate hydroxyl radicals. The results showed that the flavonoid content under the control was significantly higher than that under the 23y-CC treatment for each developmental stage except T2. The flavonoid content under the 23y-CC treatment was decreased by 17.47%, 7.11%, 20.67%, and 29.26% in Stages T1, T3, T4, and T5, respectively. The flavonoid content under the control was relatively stable from T1 to T4, but substantially decreased in the T5 stage: the flavonoid content in the T5 stage was decreased by 43.29% compared to the maximum content at the T2 stage under the control. Under the 23y-CC treatment, the flavonoid content increased quickly from T1 to T2, then showed a decreasing trend: the flavonoid content at Stage T5 was decreased by 64.27% compared to that at T2 (Figure 6a).

Tannins are a type of important plant natural antioxidant. The results showed that the tannin content under the control was significantly higher than that under the 23y-CC treatment in each developmental stage. The tannin content under the 23y-CC treatment decreased by 95.07%, 76.34%, 86.12%, 92.79%, and 100.00% compared to the control from T1 to T5, respectively. The tannin content under the control generally showed a decreasing trend from T1 to T5, and the tannin content at the T5 stage was decreased by 59.86% as compared to the T1 stage. Under the 23y-CC treatment, the tannin content increased from T1 to T2 and then decreased. No tannin content was detected in the T5 stage under the 23y-CC treatment (Figure 6b). 

### 2.5. Alterations of Protein Oxidation under Different Continuous Cropping Treatment

Sulfydryl compounds include the glutathione thiol group and protein-sulfydryl types and play important roles in cleaning up radicals. The total sulfydryl (T-SH) content results indicated significant differences between the control and 23y-CC treatments in the T3 stage. The T-SH content under the control was decreased from T3 to T5 by 83.20%, 87.50%, and 94.72% compared to the 23y-CC treatment, respectively. The T-SH content under the control increased from T1 to T2 and then decreased. Unlike under the control, the T-SH content under 23y-CC increased from T1 to T3; however, the increment from T2 to T3 was very fast, reaching a difference of 5.15 times. The T-SH content from T3 to T5 showed a slight increase and then decreased, but the decrement was only 15.31% between T3 and T5 (Figure 7a).

The amino acid side chains of proteins can be oxidized under oxidative stress conditions, which are called protein carbonyls (PCs) and may affect the structure and function of proteins. The results showed that the PC level under the control was significantly higher than that under the 23y-CC treatment from T2 to T5, while a reverse trend was observed for the T1 stage. The PC content from T2 to T5 under the 23y-CC treatment was decreased by 76.27%, 71.67%, and 67.35% compared to the control, respectively. The PC content exhibited two rapidly increasing phases (from T1 to T2 and T3 to T4) and two decreasing phases (from T2 to T3 and T4 to T5) under the control. The maximum PC content was found in the T4 stage, which was increased by 25.23 times compared to that in the T1 stage under the control. For the 23y-CC treatment, the PC content from T1 to T3 was relatively stable and peaked at T4 as well. The PC content at T5 was decreased by 82.69%, compared to the T4 stage (Figure 7b). 

### 2.6. Metabolomics Analysis 

In addition to physiological analysis of the antioxidative profile, a metabolomics analysis was further performed. The results of the Venn diagram showed that there were 116, 11, and 462 distinctive metabolites in the comparison between the control and 23y-CC treatments for Stages T1, T2, and T3, respectively (Figure 8a). The results of the volcano plot indicated that 93 and 79 metabolites were up- and down-regulated at the T1 stage in the comparison between the control and 23y-CC treatments, respectively. However, very few metabolites were detected in Stage T2, with only eight and five metabolites in the up- and down-regulation modes, respectively. The up- and down-regulated metabolites were considerably increased again at the T3 stage, with 340 and 180 metabolites, respectively (Figure 8b–d). 

The results of metabolite enrichment analysis showed that seven groups of metabolites were enriched. The first type was related to antioxidant metabolism, involved in the regulation of the redox of ROS. The detected metabolites included flavonoids, cyanoamino acid, betalain, and gingerol. The second type was related to ROS production and detoxification during photosynthesis, such as porphyrin metabolism and carotenoid biosynthesis. The third type was associated with ROS production and elimination during the electron transport chain during vitamin metabolism, including riboflavin metabolism and thiamine metabolism. The fourth type was xanthines, such as caffeine metabolism, which produce and scavenge ROS. The fifth type was the sulfur relay system, as sulfydryl compounds play important roles in the regulation of redox. The sixth type was fatty acid metabolism, such as fatty acid degradation and metabolism, especially at the late stage of development, which is closely correlated with lipid peroxidation. The seventh type was GSH and AsA metabolism, which act to eliminate ROS (Figure 8e–g). 

## 3. Discussion

Qingke is an essential staple for people living in the Qinghai–Tibet Plateau; therefore, improving Qingke yield is one of the most important goals in its production. Due to the low temperatures from late autumn to early spring in this area, the growth mode of Qingke is continuous cropping with the spring type. However, regardless of different responses to continuous cropping in various plant species, the effects of the continuous cropping of Qingke on its yield and growth have not yet been reported. 

Under continuous cropping, the yield of Qingke decreased significantly. Although some studies have reported no marked yield reduction in continuous wheat cropping [30], substantial investigations have revealed that the yield of wheat and barley decreased after continuous cropping in different regions [31,32]. Furthermore, there are various reasons for decreases in crop yield under continuous cropping, such as broken soil–micro-organism balance, deterioration of soil nutrients, and accumulated viruses resulting in plant diseases [33,34,35]. However, a systematic survey on redox regulation associated with continuous cropping has been neglected during past decades, involving the simple measurement of antioxidant enzymes such as SOD and CAT [27,36]. In the current study, the ROS profile (including ROS production and elimination) was analyzed in order to elucidate the reasons for the Qingke yield reduction from the perspective of modulation of the antioxidant defense. The results showed that the SOA production in Qingke plants under the 23y-CC treatment was significantly higher than that in control plants at all development stages, revealing that long-term continuous cropping resulted in a higher yield of SOA. Previous investigations have reported that plant ROS significantly increased under continuous cropping. For example, leaves and roots under one- and two-continuous cropping at different months during the growth of *Pinellia ternata* (Thunb.) Breit. presented significantly higher ROS than first-year planting [37]. However, detailed information on changes in different types of ROS was missing. There are multiple pathways of SOA production in plant cells, such as xanthine metabolism in the peroxisome, photosynthesis in chloroplasts, and the electron transport chain in mitochondria [38]. It is difficult to determine which pathway contributed most in the current study and it could be inferred, through both physiological and metabolomics analyses, that it co-functions through those pathways. In our results, we found that many antioxidants were enriched, as mentioned above. For example, two metabolites were related to porphyrin metabolism and another two to carotenoid biosynthesis in chloroplasts. One metabolite of caffeine metabolism, xanthine, was enriched in peroxisomes. Another phenomenon that was observed was the increase in SOA from T1 to T5 under both the control and 23y-CC treatments. These results suggest that the senescence of plants plays an important role in the production of SOA, in accordance with previous studies [39]. The increase in SOA at later stages of development was also attributed to the reduction in SASC. A higher SASC is considered to benefit the metabolism from SOA to hydrogen peroxide [40]. Considerable SOA accumulation was an important reason for the reduction in Qingke yield under continuous cropping, which was proven in previous investigations on the adverse effects of ROS stress [41,42]. In this study, the higher SASC did not necessarily result in higher hydrogen peroxide due to higher SOA content under continuous cropping. In fact, only T1, T2, and T5 exhibited a significantly higher hydrogen peroxide content under continuous cropping. The results suggest that the accumulation of hydrogen peroxide content in the cells of Qingke is a complex process. The reason may be associated with the equilibrium of biosynthesis, as well as the catalysis and elimination of hydrogen peroxide in the cells of Qingke. The metabolism of SOA conversion into hydrogen peroxide is related to an important enzyme, SOD [43]. However, in the current study, significantly lower SOD activity was found under the 23-CC treatment compared to the control, suggesting that the SOA removal ability was weaker under continuous cropping, resulting in the accumulation of SOA. Conversely, the higher activity of SOD under the control—especially in the early stages—could remove SOA in a timely manner, which should be one of the most important reasons for the lower SOA level. 

Hydrogen peroxide is another oxygen free radical. The dynamics of hydrogen peroxide content were found to be complicated, as analyzed above. In the biosynthesis of hydrogen peroxide, several pathways can contribute to its content. For example, the reaction of SOA produces hydrogen peroxide, which is catalyzed by SOD. However, the lower activity of SOD under the 23y-CC treatment was discovered first, and other enzyme/non-enzyme compounds are involved in this reaction. For example, sulfydryl of GSH can react with SOA to produce hydrogen peroxide [44]. Second, there are other pathways that produce hydrogen peroxide. Therefore, we measured the GOD and PAO activities, as hydrogen peroxide is produced as glucose and polyamines are oxidized [45,46]. Interestingly, significantly higher activities of both enzymes under the 23y-CC treatment were observed at early to middle developmental stages such as T2 and T3 when compared to the control. This result clearly shows that a higher hydrogen peroxide might be deposited during the oxidization of glucose and polyamines. As a result, once those ROS are produced, they will attack unsaturated fatty acids on the plasma membrane if they cannot be removed in a timely manner [47]. In the present study, membrane injury occurred at a late developmental stage due to the higher level of LPO under the 23y-CC treatment. Another piece of evidence was the higher MDA content under the 23y-CC treatment from middle to late developmental stages, as MDA is an important product of LPO. Continuous cropping caused LPO and led to the accumulation of MDA in cells, in agreement with many previous reports [48]. Our metabolomics results also provided powerful evidence that many metabolites related to fatty acid metabolism were enriched to protect from injury related to ROS stress under the control. Fatty acids are important in the composition of plant cell membranes, and the stability of fatty acid membranes is helpful in keeping cells healthy [49]. Thus, insufficient fatty acid metabolism indicated that membrane lipids were oxidized under continuous cropping. Therefore, the ability to clean up those ROS is a vital guarantee to maintain healthy growth in Qingke. Unfortunately, our results showed that the hydroxyl radical scavenging ability under the 23y-CC treatment was lower than that under the control. This result revealed the reason for the LPO under the 23y-CC treatment in the middle and late developmental stages [50,51]. 

As the production of hydrogen peroxide in other pathways was high under the 23y-CC treatment at the early developmental stages, the results suggested that the catalysis of hydrogen peroxide under continuous cropping was weak. In an enzymatic system, both CAT and POD can catalyze hydrogen peroxide into water [52,53]. However, the higher activity of CAT in Stages T1 and T2 and POD in Stage T2 under the 23y-CC treatment did not lead to a lower hydrogen peroxide content, which meant that the accumulation of hydrogen peroxide by various pathways was stronger than its catalysis. Under the control, lower CAT and POD in Stages T1 and T2 resulted in a lower hydrogen peroxide content. Furthermore, another enzyme, TPX, can oxidize 1,4-dithiothreitol (DTT) to eliminate hydrogen peroxide in plant cells [54]. Its higher activity in the early stages under the control helped to reduce the accumulation of hydrogen peroxide content. In order to balance the hydrogen peroxide content, two non-enzymatic systems were also assayed. The glutathione system plays an important role in redox regulation in plant cells [55]. In the current study, the higher GSH content under the control revealed the weak ROS scavenging capacity when Qingke was under the continuous cropping condition. This is one of the main important reasons why the antioxidant defensive system lowered the resulting grain yield under continuous cropping. Not only the GSH content but also the GSSG content was higher under the control, suggesting a high reduction in activity from GSH. However, the GSH content was far higher than that of GSSG under both treatments. The difference reached 393 to 132,130 times during the development stages of Qingke. The results revealed that it was necessary for the plants to maintain higher GSH levels for their healthy growth [56]. The redox status is regulated by many enzymes in this system. Although a very high activity of GCL was observed under the 23y-CC treatment, indicating more GSH biosynthesis was required for the detoxification of ROS, the lower content of GSH under continuous cropping suggests that a higher consumption of GSH took place in the plant cells. The direct evidence comprised the very high activity of GPX from Stages T1 to T3, which functions in the oxidation of GSH. For the control, a higher activity of GST and GPX at the early and late developmental stages, respectively, will also oxidize GSH, which potentially reduces the GSH content. However, the higher GSH content than that under the 23y-CC treatment at all stages is correlated with the high activity of two enzymes, GR and TrxR. Both enzymes showed very high activity at the early developmental stages and could reduce GSSG to regenerate GSH. Thus, the oxidized GSH by GST and GPX can offset the reverse reduction to GSH by GR and TrxR to the supplementary GSH in Qingke cells. This might be the main reason for the lower content of GSH under continuous cropping. Maintaining a higher GSH content is beneficial to cleaning up hydrogen peroxide; therefore, lower hydrogen peroxide content was detected during the early development stages of Qingke under the control treatment. 

In addition to GSH, the AsA system also plays an important role in the redox balance [20,57]. One of the main functions of AsA is to remove hydrogen peroxide [58,59]. The results showed that the considerably lower AsA content in the early developmental stages under continuous cropping resulted in the accumulation of hydrogen peroxide. Under continuous cropping, the lower AsA content was proven according to two pieces of evidence. First, a key enzyme, GalLDH, plays a key role in the biosynthetic pathway of AsA, which was decreased under continuous cropping [60]. Thus, the AsA content was low under the 23y-CC treatment, especially in the early developmental stages. Second, AsA could be strongly oxidized by APX, due to its higher activity under continuous cropping in late developmental stages. Another enzyme, AAO, showed a higher content during the late developmental stages, which can also oxidize AsA. Therefore, both APX and AAO resulted in a lower AsA content. Once AsA is oxidized, the product DHA can be reversibly converted into AsA again [23], catalyzed by MDHAR. The current study found that the MDHAR activity under continuous cropping was very low in all developmental stages, suggesting that the amount of DHA that can be reduced into AsA should be very low. Therefore, the deficiency of AsA led to the accumulation of hydrogen peroxide under continuous cropping and, hence, the injury of the cell membrane (i.e., LPO).

In addition to GSH and AsA, other antioxidants also alleviate ROS stress. In this study, the total antioxidant capacity was reduced under continuous cropping. The results indicate that the content of some antioxidants might be affected by continuous cropping. Our results showed that flavonoid and tannin contents were decreased under continuous cropping, which was an important reason for its low antioxidant capacity [61]. Our metabolomics results also showed that many other metabolites—including flavonoids, flavone, cyanoamino acid, betalain, and gingerol—were enriched. Thus, substantial antioxidants were activated to prevent ROS stress under normal growth conditions, while the opposite resulted in the suppression of development in Qingke under continuous cropping. In Qingke cells, a very high T-SH content was detected in the middle to late developmental stages under continuous cropping. As the sulfydryl of GSH was low under continuous cropping, the protein sulfydryl content under continuous cropping should be much higher, which is beneficial in repairing oxidized carbonyl proteins. The result of a low PC level provided powerful evidence for the mentioned biological process under continuous cropping.

## 4. Materials and Methods

### 4.1. Plant Materials and Experimental

A field experiment was carried out at the experimental station of Tibet Academy of Agricultural and Animal Husbandry Sciences, N 29°56′ and E 91°7′, where the altitude is 3794.0 m. The Qingke variety used in the study was Zangqing 2000. The previous crop under 23-year continuous cropping (23y-CC) was Qingke, while a rotation of rapeseed and pea took place before planting Qingke under the control. The block area was 100 m^2^. The Qingke seeds were manually direct-seeded at a density of 225 kg ha^−1^. The fertilizer was applied at a ratio of 22:10:13 for N:P_2_O_5_:K_2_O in the same amount at each treatment and each year. There was no irrigation during the experiment. During the growth period from March to August, both the highest and lowest temperature showed an increasing trend from March to June and then kept very small variations. However, for precipitation, it was low from March to May and then increased sharply to June, where it reached 86.9 mm (Figure 9).

### 4.2. Experimental Design and Sampling

The experimental design was a randomized complete block design with three replications (n = 3). The treatment was composed of 23-year continuous Qingke cropping and a control (the first year of planting of Qingke). Crop samples were taken at five developmental stages, including the seedling, tillering, jointing, flowering, and seed-filling stages, which were designated as T1, T2, T3, T4, and T5, respectively. One-third of each area was used for yield determination, and two-thirds were used for sampling. During sampling, the plants were randomized and collected in the core area, excluding border plants, to avoid marginal effects. One piece of each leaf sample was conserved in liquid nitrogen, and another piece was immediately taken directly to the lab. The samples were saved in liquid nitrogen in a −80 °C refrigerator to measure enzymatic activities. Additionally, the fresh samples were killed at 100 °C for 30 min and then dried at 70 °C for the determination of some physiological indices. 

### 4.3. ROS Production and Analysis of Scavenging Capacity 

For histochemical staining, leaf tissues were cut into small pieces, followed by staining, incubation, bleaching, and imaging of O_2_^•−^ and H_2_O_2_, which were performed according to Ueda et al. (2013) [62] and Meena et al. (2016) [63], respectively. The superoxide anion (SOA) and hydrogen peroxide were assayed using a testing kit (Suzhou Coming Biotechnology Co., Ltd. Suzhou, China). Briefly, for O_2_^•−^ content, 1.0 g sample was homogenized with 5.0 mL of phosphate buffer solution (PBS, pH7.8) on ice. The homogenate was volumed up to 10 mL by PBS and then centrifugated 10 min at 8000 rpm/min, 4 °C. A total of 2.0 mL of supernatant was taken into a tube, and then 1.5 mL PBS and 0.5 mL 10 mmol L^−1^ hydroxylamine hydrochloride were added. The mixture was placed in a water bath for 20 min at 25 °C. A total of 2.0 mL of the reaction solution was taken into another tube, and then 2.0 mL of 17 mmol L^−1^ p-aminobenzene sulfonic acid solution and 2.0 mL of 7 mmol L^−1^ α-naphthylamine solution were added and mixed. The mixture was reacted in a water bath for 30 min at 30 °C. When the reaction was finished, the content was measured at 530 nm using ddH_2_O as a control in an ultraviolet spectrophotometer (UV-2550, Shimadzu, Kyoto, Japan). For H_2_O_2_ content, 1.0 g sample was homogenized with 3 mL of pre-cooled acetone (4 °C) on ice. The homogenate was made up to a volume of 5 mL with pre-cooled acetone and then centrifugated for 10 min at 8000 rpm/min, 4 °C. A total of 1 mL of supernatant was taken into a tube, and then 0.1 mL of 5% TiSO_4_ solution and 0.2 mL of ammonium hydroxide were added. When sediment appeared in the mixture, 5 mL of 2 mol L^−1^ H_2_SO_4_ solution was added into this mixture to dissolve the sediment completely. The reaction solution was used for H_2_O_2_ content determination at 410 nm using ddH_2_O as a control in an ultraviolet spectrophotometer (UV-2550, Shimadzu, Japan).

The ROS scavenging capacity, including hydroxyl radical scavenging ability (HRSA), superoxide anion scavenging capacity (SASC), and total antioxidant capacity (T-AOC) with 2,2-Diphenyl-1-picrylhydrazyl (DPPH), was measured using a testing kit (Suzhou Coming Biotechnology Co., Ltd. Suzhou, China), according to the instructions.

Membrane lipid peroxidation and malondialdehyde (MDA) content analyses were conducted using a testing kit supplied by Suzhou Coming Biotechnology Co., Ltd. Suzhou, China.

### 4.4. Enzymatic ROS Metabolic Analysis

The activities of enzymes in this study included catalase (CAT), peroxidase (POD), superoxide dismutase (SOD), glucose oxidase (GOD), and polyamine oxidase (PAO). The enzymatic activities were measured using a testing kit supplied by Suzhou Coming Biotechnology Co., Ltd. Suzhou, China.

### 4.5. Glutathione Metabolism Analysis

Glutathione metabolism was assayed according to indices including reduced glutathione (GSH) and oxidized glutathione (GSSG), glutathione peroxidase (GPX), glutathione S-transferase (GST), glutathione reductase (GR), thioredoxin peroxidase (TPX), glutamate cysteine ligase (GCL), and thioredoxin reductase (TrxR), which were measured using a testing kit supplied by Suzhou Coming Biotechnology Co., Ltd. Suzhou, China.

### 4.6. Ascorbic Acid Metabolism Analysis

The ascorbic acid metabolism was analyzed according to indices including reduced ascorbic acid (AsA), dehydroascorbate (DHA), dehydroascorbate reductase (DHAR), ascorbate peroxidase (APX), monodehydroascorbate reductase (MDHAR), ascorbate oxidase (AAO), and L-galactono-1,4-lactone dehydrogenase (GalLDH). The enzymatic activities were assayed using a testing kit supplied by Suzhou Coming Biotechnology Co., Ltd. Suzhou, China.

### 4.7. Other Antioxidants Analysis

Other contents of important antioxidants were assayed, including flavonoids, tannins, total sulfydryl (T-SH) content, and protein carbonyl (PC). The measurement followed the instructions of a testing kit supplied by Suzhou Coming Biotechnology Co., Ltd. Suzhou, China.

### 4.8. Metabolomics Analysis

Leaf samples were taken at seedling (T1), flowering (T3), and seed-filling (T5) stages. The extraction of metabolites, UPLC-MS/MS, data quality assessment, annotation analysis, differential expression analysis, functional enrichment, and so on, were measured according to Hao et al. (2022) [64]. |log2foldChange| > 1, *p*-value < 0.05, and variable importance in projection (VIP ≥ 1) were determined to indicate differentially accumulated metabolites (DAMs). Briefly, UPLC-MS/MS analyses were performed using a Vanquish UPLC system (ThermoFisher, Germany) coupled with an Orbitrap Q Exactive TM HF mass spectrometer or Orbitrap Q Exactive TMHF-X mass spectrometer (Thermo Fisher, Germany) in Novogene Co., Ltd. (Beijing, China). Samples were injected onto a Hypersil Goldcolumn (100 × 2.1 mm, 1.9 μm) using a 12 min linear gradient at a flow rate of 0.2 mL/min. The eluents for the positive polarity mode were eluent A (0.1% FA in water) and eluent B (methanol). The eluents for the negative polarity mode were eluent A (5 mM ammonium acetate, pH 9.0) and eluent B (methanol). The solvent gradient was set as follows: 2% B, 1.5 min; 2–85% B, 3 min; 85–100% B, 10 min; 100–2% B, 10.1 min; 2% B, 12 min. A Q Exactive TM HF mass spectrometer was operated in positive/negative polarity mode with a spray voltage of 3.5 kV, capillary temperature of 320 °C, sheath gas flow rate of 35 psi and aux gas flow rate of 10 L/min, S-lens RF level of 60, and aux gas heater temperature of 350 °C.

### 4.9. Statistics

Data are expressed as the mean of three replicates (n = 3), and the analysis was performed using the IBM SPSS v.22.0 statistical software. Duncan’s multiple range test (DMRT) was conducted to evaluate significant treatment effects and time points (Appendix A) at the significance level of *p* ≤ 0.05. 

## 5. Conclusions

In the current study, a significant reduction in grain yield in Qingke under the 23y-CC treatment was obtained. The decrease in grain yield under continuous cropping was associated with ROS stress during each main developmental stage of the Qingke plants. The increase in ROS, including SOA at all stages and hydrogen peroxide at early stages, was the main reason for membrane LPO under continuous cropping. Higher SASC did not reduce SOA content, but could increase the hydrogen peroxide content under continuous cropping. The elevated hydrogen peroxide under continuous cropping at early developmental stages was the result of combined functions, including various pathways and hydrogen peroxide production, for example, due to factors such as higher GOD and PAO activity and conversion by GSH and SOD. The lower content of GSH under continuous cropping was due to the low regeneration of GSH from GSSG. Furthermore, lower AsA content under continuous cropping was observed due to the inhibition of AsA biosynthesis caused by lower GalLDH activity. The higher oxidation of AsA and the lower reduction in DHA under continuous cropping were other important reasons for the lower AsA level. Our study provides new insights into regulating Qingke grain yield through the reduction in ROS stress by increasing GSH and AsA contents in the context of continuous cropping.

## Figures and Tables

**Figure 1 plants-13-01003-f001:**
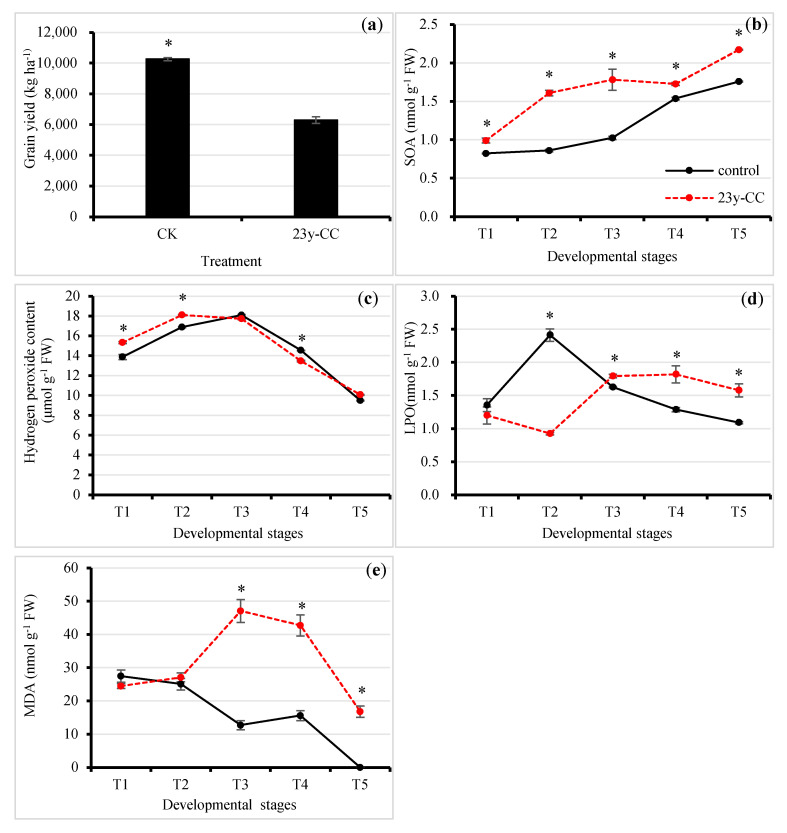
The effect of continuous cropping on (**a**) grain yield; (**b**) superoxide anion level; (**c**) hydrogen peroxide content; (**d**) lipid peroxidation level; and (**e**) malondialdehyde content from Stages T1 to T5. (**f**,**g**) NBT and DAB staining in leaves from T1 to T5 between the control and 23y-CC. T1–T5 indicate the seedling, tillering, jointing, flowering, and seed-filling stages, respectively. “*” indicates significant differences between treatments using Duncan’s method (*p* < 0.05). Error bars indicate standard deviation (n = 3). DAB, diaminobenzidine; MDA, malondialdehyde; NBT, nitrotetrazolium blue chloride; LPO, lipid peroxidation; SOA, superoxide anion; 23y-CC, 23-year continuous cropping.

**Figure 2 plants-13-01003-f002:**
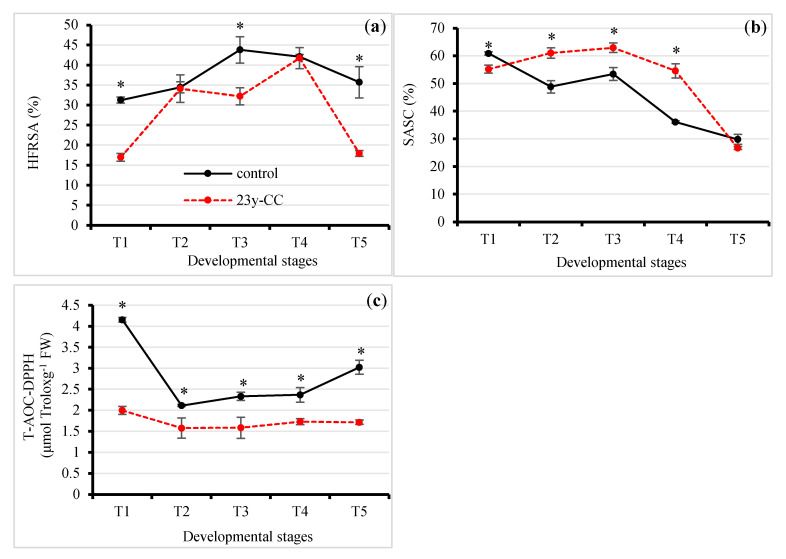
Effect of continuous cropping on (**a**) HRSA, (**b**) SASC, and (**c**) T-AOC-DPPH from Stages T1 to T5. T1–T5 indicate the seedling, tillering, jointing, flowering, and seed-filling stages, respectively. “*” indicates a significant difference between treatments using Duncan’s method (*p* < 0.05). Error bars indicate standard deviation (n = 3). HRSA, hydroxyl free radical scavenging ability; SASC, superoxide anion scavenging capacity; T-AOC-DPPH, total antioxidant capacity with 2,2-Diphenyl-1-picrylhydrazyl; 23y-CC, 23-year continuous cropping.

**Figure 3 plants-13-01003-f003:**
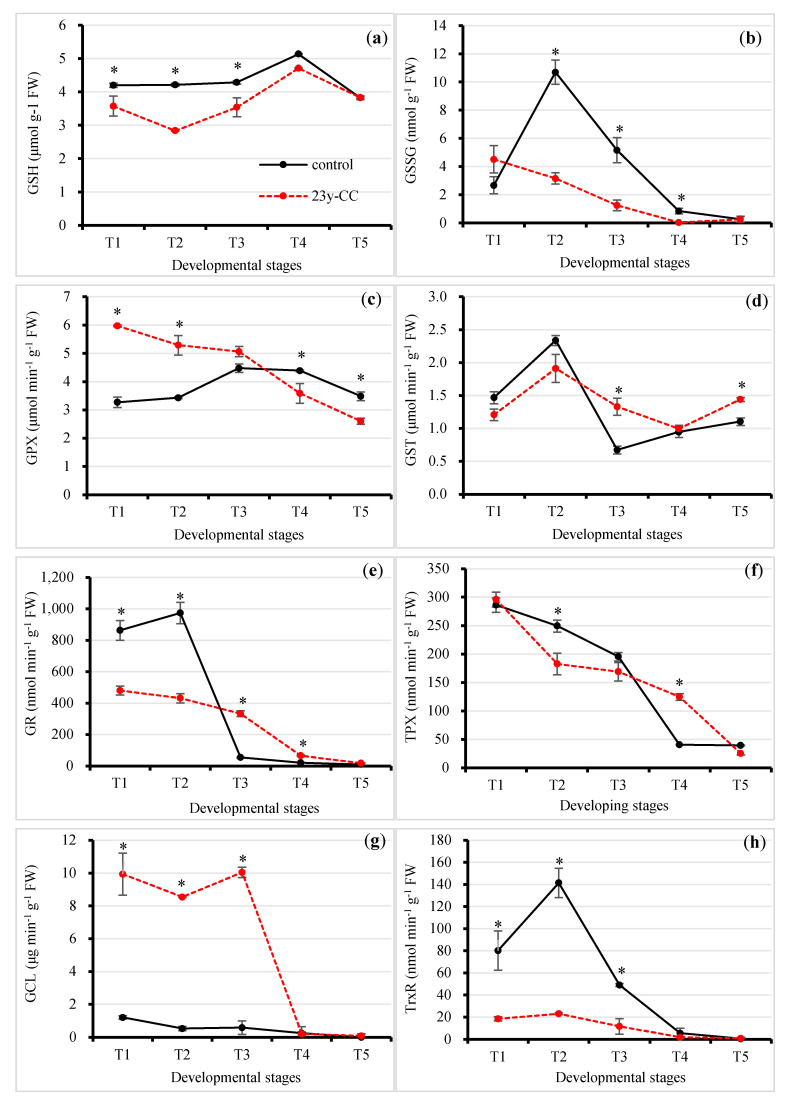
Effect of continuous cropping on glutathione metabolism: (**a**) GSH content; (**b**) GSSG content; (**c**) GPX activity; (**d**) GST activity; (**e**) GR activity; (**f**) TPX activity; (**g**) GCL activity; (**h**) TrxR activity from stages T1 to T5. T1–T5 indicate seedling, tillering, jointing, flowering, and seed-filling stage, respectively. “*” indicates a significant difference between treatments using Duncan’s method (*p* < 0.05). Error bars indicate standard deviation (n = 3). GCL, glutamate cysteine ligase; GPX, glutathione peroxidase; GR, glutathione reductase; GSH, reduced glutathione; GSSG, oxidized glutathione; GST, glutathione S-transferase; TPX, thioredoxin peroxidase; TrxR, thioredoxin reductase; 23y-CC, 23-year continuous cropping.

**Figure 4 plants-13-01003-f004:**
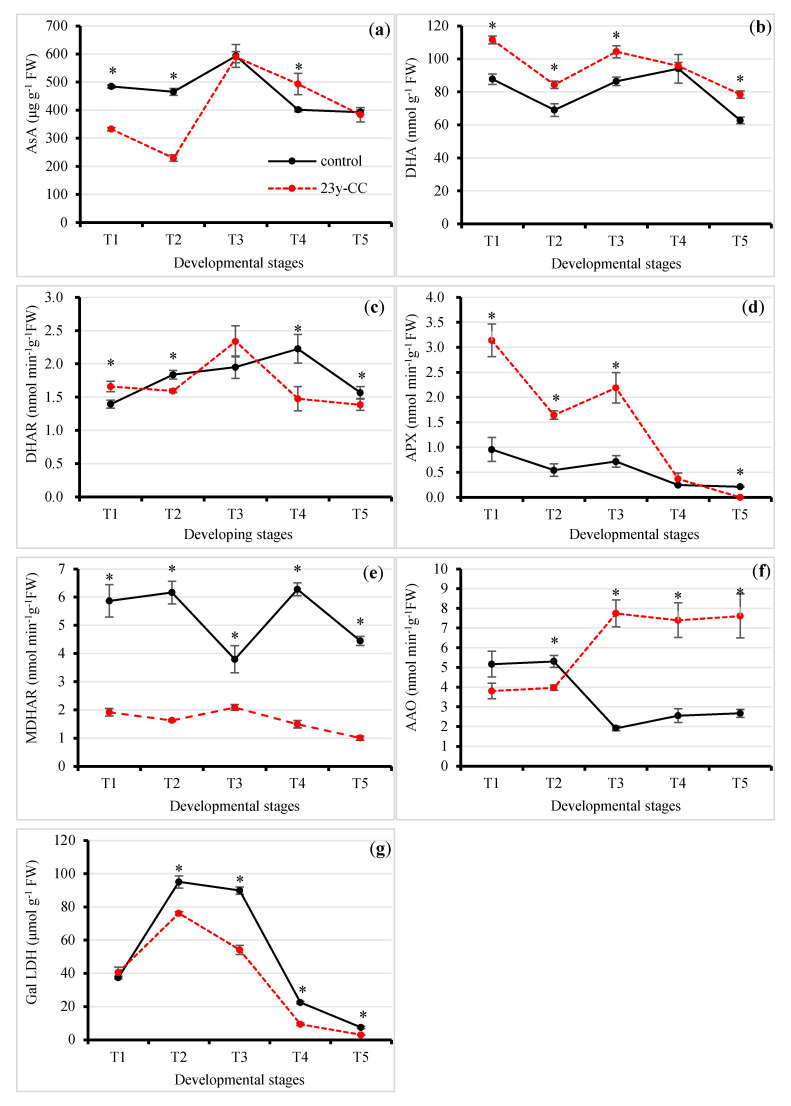
Effect of continuous cropping on ascorbic acid metabolism: (**a**) AsA content; (**b**) DHA content; (**c**) DHAR activity; (**d**) APX activity; (**e**) MDHAR activity; (**f**) AAO activity; (**g**) GalLDH activity from stages T1 to T5. T1–T5 indicate seedling, tillering, jointing, flowering, and seed-filling stages, respectively. “*” indicates a significant difference between treatments using Duncan’s method (*p* < 0.05). Error bars indicate standard deviation (n = 3). AAO, ascorbate oxidase; APX, ascorbate peroxidase; AsA, reduced ascorbic acid; DHA, dehydroascorbate; DHAR, dehydroascorbate reductase; GalLDH, L-galactono-1,4-lactone dehydrogenase; MDHAR, monodehydroascorbate reductase; 23y-CC, 23-year continuous cropping.

**Figure 5 plants-13-01003-f005:**
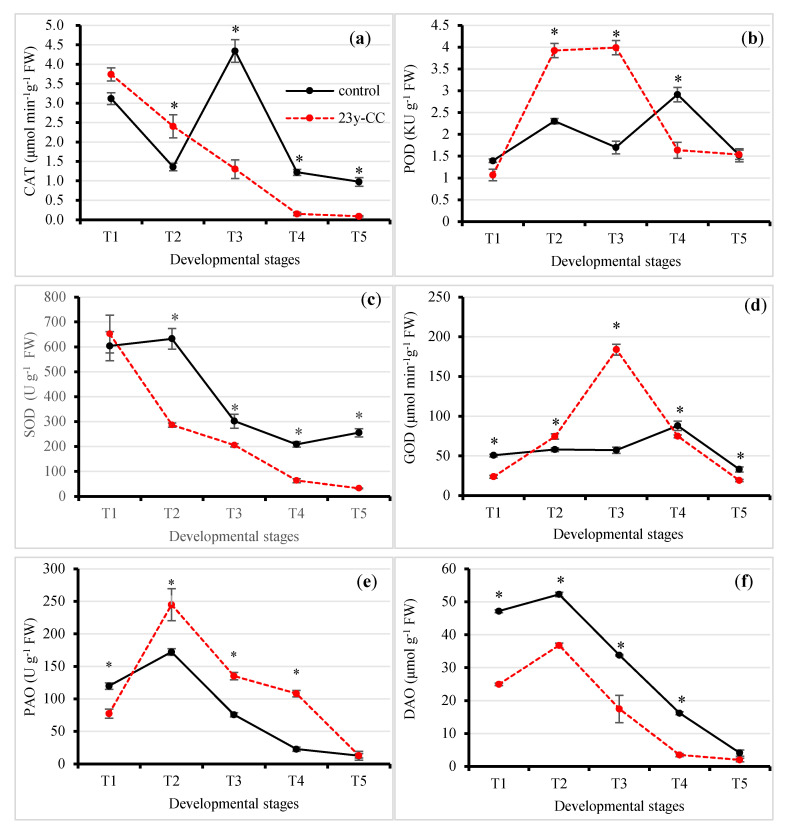
Effect of continuous cropping on antioxidant enzyme activities from Stages T1 to T5: (**a**) CAT, (**b**) POD, (**c**) SOD, (**d**) GOD, (**e**) PAO, and (**f**) DAO. T1–T5 indicate the seedling, tillering, jointing, flowering, and seed-filling stages, respectively. “*” indicates a significant difference between treatments using Duncan’s method (*p* < 0.05). Error bars indicate standard deviation (n = 3). CAT, catalase; DAO, diamine oxidase; GOD, glucose oxidase; PAO, polyamine oxidase; POD, peroxidase; SOD, superoxide dismutase; 23y-CC, 23-year continuous cropping.

**Figure 6 plants-13-01003-f006:**
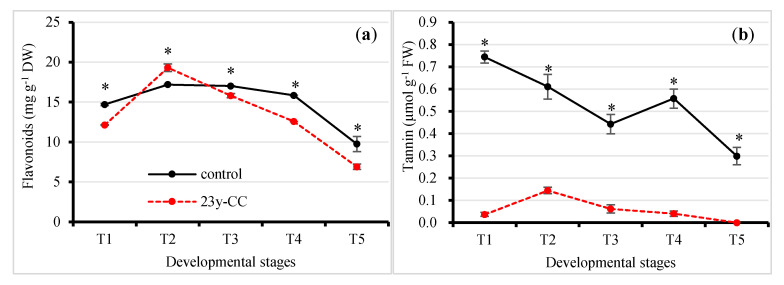
Effect of continuous cropping on antioxidant contents from Stages T1 to T5: (**a**) flavonoids, (**b**) tannins. T1–T5 indicate the seedling, tillering, jointing, flowering, and seed-filling stages, respectively. “*” indicates a significant difference between treatments using Duncan’s method (*p* < 0.05). Error bars indicate standard deviation (n = 3). 23y-CC, 23-year continuous cropping.

**Figure 7 plants-13-01003-f007:**
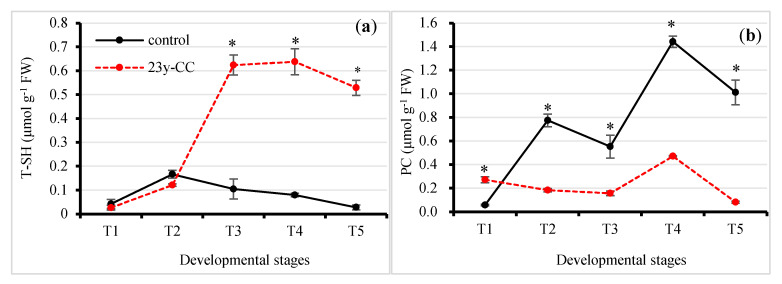
Effect of continuous cropping on (**a**) T-SH content and (**b**) PC level from Stages T1 to T5. T1–T5 indicate the seedling, tillering, jointing, flowering, and seed-filling stages, respectively. “*” indicates a significant difference between treatments using Duncan’s method (*p* < 0.05). Error bars indicate standard deviation (n = 3). PC, protein carbonyl; T-SH, total sulfydryl; 23y-CC, 23-year continuous cropping.

**Figure 8 plants-13-01003-f008:**
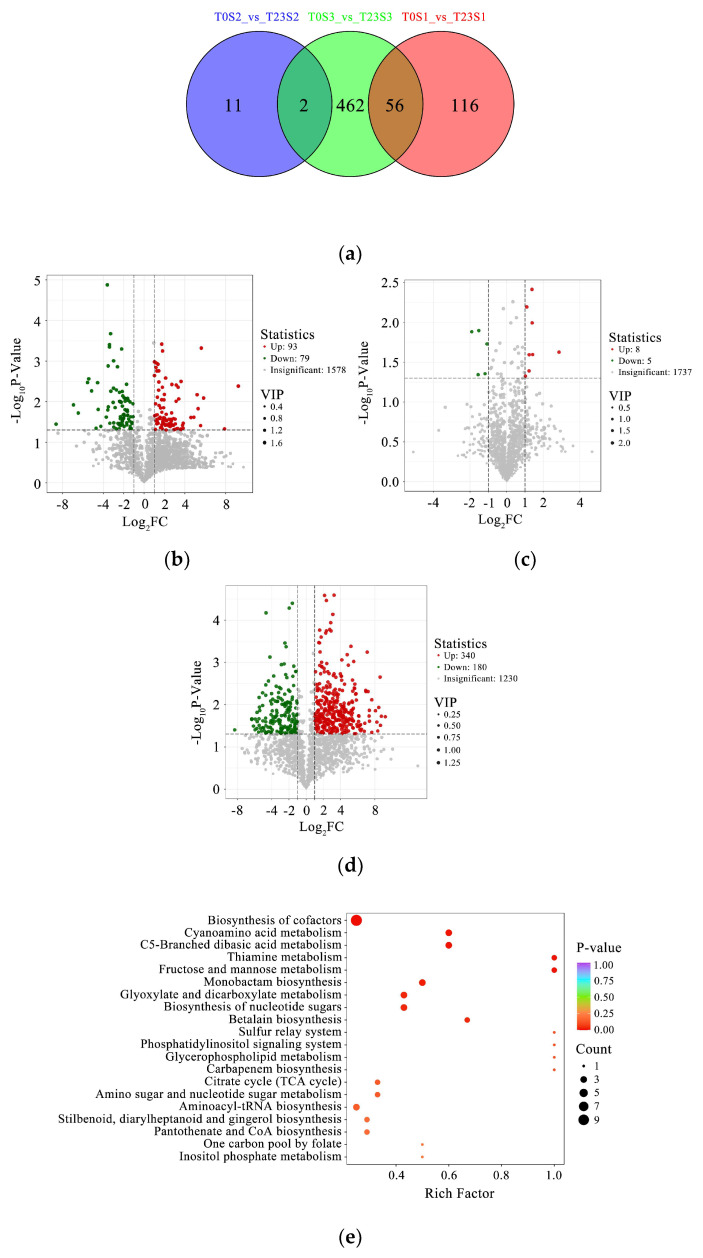
Metabolomics analysis under 23-year continuous cropping and control treatments: (**a**) Venn diagram of T0S1 vs. T23S1, T0S2 vs. T23S2, and T0S3 vs. T23S3; (**b**–**d**) volcano diagrams of T0S1 vs. T23S1, T0S2 vs. T23S2, and T0S3 vs. T23S3 with differentially accumulated metabolites. (**e**–**g**) Rich factor analysis of T0S1 vs. T23S1, T0S2 vs. T23S2, and T0S3 vs. T23S3. S1, seedling stage; S2, jointing stage; S3, seed-filling stage; T0, control; T23, 23-year continuous cropping.

**Figure 9 plants-13-01003-f009:**
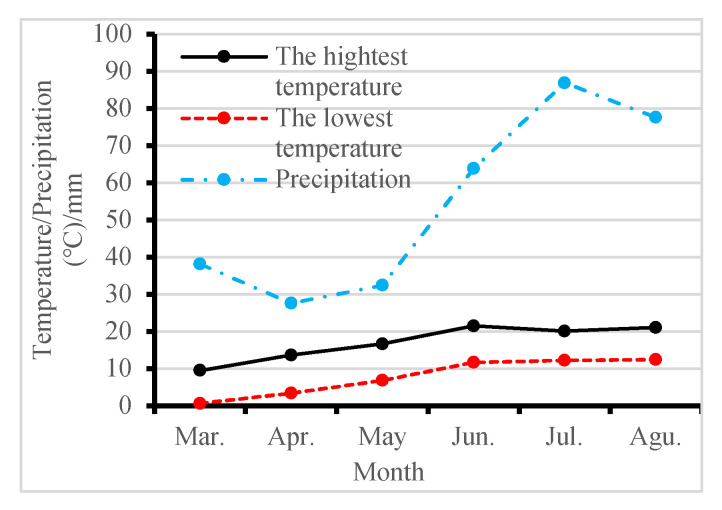
Weather conditions including the highest and lowest temperatures and precipitation during Qingke growth periods.

## Data Availability

Data are available by reasonable request from the corresponding author.

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
