# Peer review of "Elevated ROS Levels Caused by Reductions in GSH and AsA Contents Lead to Grain Yield Reduction in Qingke under Continuous Cropping"

_plants, 2024, doi:10.3390/plants13071003_

Round 1

Reviewer 1 Report

Comments and Suggestions for Authors

The manuscript deals with the assessment of antioxidant response of qingke to continuous cropping system. Authors performed multifactorial metabolomic study with a wide range of antioxidants tested. The paper is generally well written and the results are described in detail. However, the paper has some flaws. The details are listed below:

Abstract should be shortened

L59-60: highlight the ubiquitous role of antioxidant enzymes in mitigating different abiotic stresses including pesticides, heavy metals, drought, etc. which generate ROS. For this purpose refer to https://doi.org/10.1016/j.chemosphere.2022.136284

L519: increase the dimensions of the Figure

L678: indicate weather conditions

L699: briefly describe determination of O2- and H2O2

L734: indicate which metabolites were tested. Add the procedure of their extraction and UPLC-MS/MS determination

Comments on the Quality of English Language

Moderate editing of English language required

Reviewer 2 Report

Comments and Suggestions for Authors

The manuscript is dedicated to the studies of the effect of continuous cropping on the ROS stress and yield of Qingke barley. A wide variety of parameters was tested.

The main concern is how different the soil type was initially, before setting up the experiment.  The soil on different slopes could have been initially different before the start of the experiment, which could have resulted in the differences between the parameters. As far as I understand, there were two fields - one after 23 years of cropping and another one after a rotation of growing rapeseed and pea. Of course, different plant species can have different preferences towards the elements of mineral nutrition, which will lead to depletion of different macro- and micro-nutrients, and they can differently affect the soil micro biome etc. via root exudation and leaf litter etc. So, to understand what lies behind the differences in ROS stress, it is important to know the difference between soil composition and the pH.

 Figure 1 - NBT and DAB staining in leaves is not clearly visible. Please enlarge the photos.

Figure 1c - at T3 and T5 stages, isn`t there a misprint, should there be  asterisks? It looks like there is no difference between the variants.

Line 233 - please correct GSSS for GSSG

In the text it is written "For thioredoxin peroxidase (TPX) activity, the significant differences were found at T2 and T4 stages." However, in figure 3f stages T3 and T5 are also marked with asterisks. It looks like there is a mistake in the figure. Please, check for inconsistencies between the figures and description throughout the manuscript.

Lines 315-316 "Dehydroascorbate reductase (DHAR) catalyzes GSH to reduce DHA into AsA and GSSG." Please rephrase. This is unclear.

Line 391: The POD activity under control generally increased from T1 to T4 and then decreased under control." Please rephrase. This is unclear. From the figure it is unclear whether POD activity at stage T3 was significantly higher than at T1.

The authors often compare the parameters under study between the stages  T1-T5 within one variant (e.g. how they changed with time in control). However, in the plots only significant differences between control and 23y-CC are marked with asterisks. To compare the time points, the differences between the stages should be marked with lettering according to the results of corresponding statistical analysis.

Lines 394-395 "The  maximum POD activity at T4 and T3 was 1.08- and 2.60-fold of T5 under control and 23y-CC treatment, respectively." Please rephrase, this is not clear.

Lines 425-426: "Result showed that DAO activity under 23y-CC treatment was significantly higher than that under control throughout whole developing stages." - at T5 there was no difference

Lines 427-428 - (a) and (b) have been moved here from the graphs. Please correct.

Line 463 - misprint "tammins" should be corrected for 'tannins'

Figure 8 -  everything is too small and unreadable. It is impossible to see what is depicted there. Maybe it would be better to make several separate figures instead of one.

Lines 594-595 " Interestingly, significantly higher activities of both enzymes under 23y-CC treatment was found as compared to control." - GOD activity was higher only at T2 and T3 stages.

Comments on the Quality of English Language

Extensive editing of the English language is required.

Round 2

Reviewer 1 Report

Comments and Suggestions for Authors

The Authors have corrected the manuscript. I have no more comments.

Author Response

Thank you for your confirmation.

Reviewer 2 Report

Comments and Suggestions for Authors

The manuscript has been substantially revised. I have no further questions.

Comments on the Quality of English Language

Moderate editing of English language is required.

Author Response

Thank you for your confirmation. Our manuscript has been edited by English editor of the MDPI. I will upload the certification during I upload revised version of manuscript. 
